# NOISY AGENTS: SELF-SUPERVISED EXPLORATION BY PREDICTING AUDITORY EVENTS

## ABSTRACT

Humans integrate multiple sensory modalities (*e.g.*, visual and audio) to build a causal understanding of the physical world. In this work, we propose a novel type of intrinsic motivation for Reinforcement Learning (RL) that encourages the agent to understand the causal effect of its actions through auditory event prediction. First, we allow the agent to collect a small amount of acoustic data and use K-means to discover underlying auditory event clusters. We then train a neural network to predict the auditory events and use the prediction errors as intrinsic rewards to guide RL exploration. We first conduct an in-depth analysis of our module using a set of Atari games. We then apply our model to audio-visual exploration using the Habitat simulator and active learning using the ThreeDWorld (TDW) simulator. Experimental results demonstrate the advantages of using audio signals over vision-based models as intrinsic rewards to guide RL explorations.

## 1 INTRODUCTION

Deep Reinforcement Learning algorithms aim to learn a policy of an agent to maximize its cumulative rewards by interacting with environments and have demonstrated substantial success in a wide range of application domains, such as video game (Mnih et al., 2015), board games (Silver et al., 2016), and visual navigation (Zhu et al., 2017). While these results are remarkable, one of the critical constraints is the prerequisite of carefully engineered dense reward signals, which are not always accessible. To overcome these constraints, researchers have proposed a range of intrinsic reward functions. For example, curiosity-driven intrinsic reward based on prediction error of current (Burda et al., 2018b) or future state (Pathak et al., 2017) on the latent feature spaces have shown promising results. Nevertheless, visual state prediction is a non-trivial problem as visual state is high-dimensional and tends to be highly stochastic in real-world environments.

The occurrence of physical events (*e.g.* objects coming into contact with each other, or changing state) often correlates with both visual and auditory signals. Both sensory modalities should thus offer useful cues to agents learning how to act in the world. Indeed, classic experiments in cognitive and developmental psychology show that humans naturally attend to both visual and auditory cues, and their temporal coincidence, to arrive at a rich understanding of physical events and human activity such as speech (Spelke, 1976; McGurk & MacDonald, 1976). In artificial intelligence, however, much more attention has been paid to the ways visual signals (e.g., patterns in pixels) can drive learning. We believe this misses important structure learners could exploit. As compared to visual cues, sounds are often more directly or easily observable causal effects of actions and interactions. This is clearly true when agents interact: most communication uses speech or other nonverbal but audible signals. However, it is just as much in physics. Almost any time two objects collide, rub or slide against each other, or touch in any way, they make a sound. That sound is often clearly distinct from background auditory textures, localized in both time and spectral properties, hence relatively easy to detect and identify; in contrast, specific visual events can be much harder to separate from all the ways high-dimensional pixel inputs are changing over the course of a scene. The sounds that result from object interactions also allow us to estimate underlying causally relevant variables, such as material properties (*e.g.*, whether objects are hard or soft, solid or hollow, smooth, or rough), which can be critical for planning actions.

These facts bring a natural question of how to use audio signals to benefit policy learning in RL. In this paper, our main idea is to use sound prediction as an intrinsic reward to guide RL exploration.

Intuitively, we want to exploit the fact that sounds are frequently made when objects interact, or other causally significant events occur, like cues to causal structure or candidate subgoals an agent could discover and aim for. A naïve strategy would be to directly regress feature embeddings of audio clips and use feature prediction errors as intrinsic rewards. However, prediction errors on feature space do not accurately reflect how well an agent understands the underlying causal structure of events and goals. It also remains an open problem on how to perform appropriate normalizations to solve intrinsic reward diminishing issues. To bypass these limitations, we formulate the sound-prediction task as a classification problem, in which we train a neural network to predict auditory events that occurred after applying action to a visual scene. We use classification errors as an exploration bonus for deep reinforcement learning. Concretely, our pipeline consists of two exploration phases. In the beginning, the agent receives an incentive to actively collect a small amount of auditory data by interacting with the environment. Then we cluster the sound data into auditory events using K-means. In the second phase, we train a neural network to predict the auditory events conditioned on the embedding of visual observations and actions. The state that has the wrong prediction is rewarded and encouraged to be visited more. We demonstrate the effectiveness of our intrinsic module on game playing with in Atari (Bellemare et al., 2013), audio-visual exploration in Habitat (Savva et al., 2019), active learning using a rolling robot in ThreeDWorld (TDW) (Gan et al., 2020a). In summary, our work makes the following contributions:

- We introduce a novel and effective auditory event prediction (AEP) framework to make use of the auditory signals as intrinsic rewards for RL exploration.
- We perform an in-depth study on Atari games to understand our audio-driven exploration works well under what circumstances.
- We demonstrate our new model can enable a more efficient exploration strategy for audio-visual embodied navigation on the Habitat environment.
- We show that our new intrinsic module is more stable in the 3D multi-modal physical world environment and can encourage interest actions that involved physical interactions.

## 2 RELATED WORK

**Audio-Visual Learning.** In recent years, audio-visual learning has been studied extensively. By leveraging audio-visual correspondences in videos, it can help to learn powerful audio and visual representations through self-supervised learning (Owens et al., 2016b; Aytar et al., 2016; Arandjelovic & Zisserman, 2017; Korbar et al., 2018; Owens & Efros, 2018). In contrast to the widely used correspondences between these two modalities, we take a step further by considering sound as causal effects of actions.

**RL Explorations.** The problem of exploration in Reinforcement Learning (RL) has been an active research topic for decades. There are various solutions that have been investigated for encouraging the agent to explore novel states, including rewarding information gain (Little & Sommer, 2013), surprise (Schmidhuber, 1991; 2010), state visitation counts (Tang et al., 2017; Bellemare et al., 2016), empowerment (Klyubin et al., 2005), curiosity (Pathak et al., 2017; Burda et al., 2018a) disagreement (Pathak et al., 2019) and so on. A separate line of work (Osband et al., 2019; 2016) adopts parameter noises and Thompson sampling heuristics for exploration. For example, Osband et al. (2019) trains multiple value functions and makes use of the bootstraps for deep exploration. Here, we mainly focus on the problem of using intrinsic rewards to drive explorations. The most widely used intrinsic motivation could be roughly divided into two families. The first one is count-based approaches (Strehl & Littman, 2008; Bellemare et al., 2016; Tang et al., 2017; Ostrovski et al., 2017; Martin et al., 2017; Burda et al., 2018b), which encourage the agent to visit novel states. For example, Burda et al. (2018b) employs the prediction errors of a self-state feature extracted from a fixed and random initialized network as exploration bonuses and encourage the agent to visit more previous unseen states. Another one is the curiosity-based approach (Stadie et al., 2015; Pathak et al., 2017; Haber et al., 2018; Burda et al., 2018a), which is formulated as the uncertainty in predicting the consequences of the agent's actions. For instance, Pathak et al. (2017) and Burda et al. (2018a) use the errors of predicting the next state in the latent feature space as rewards. The agent is then encouraged to improve its knowledge about the environment dynamics. In contrast to previous work purely using vision, we make use of the sound signals as rewards for RL explorations.

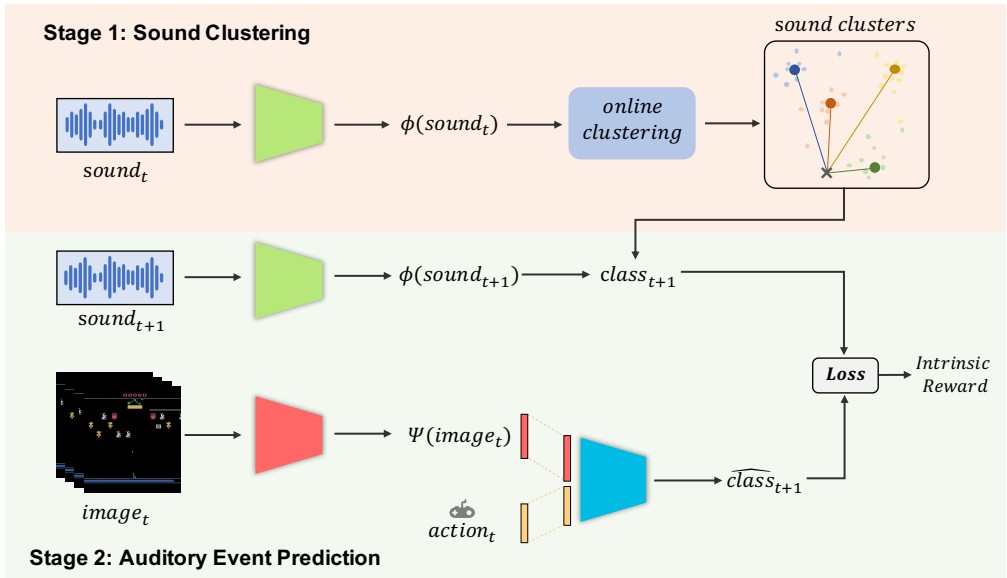

Figure 1: **An overview of our framework.** Our model consists of two stages: sound clustering and auditory event prediction. The agent starts to collect a diverse set of sound through limited environment interactions (*i.e.* 10K) and then clusters them into auditory event classes. In the second stage, the agent uses errors of auditory events predictions as intrinsic reward to explore the environment.

**Sounds and Actions.** There are numerous works to explore the associations between sounds and actions. For example, Owens et al. (2016a) made the first attempt to collect an audio-video dataset through physical interaction with objects and train an RNN model to generate sounds for silent videos. Shlizerman et al. (2018); Ginosar et al. (2019) explore the problem of predicting body dynamics from music and body gesture from speech. Gan et al. (2020b) and Chen et al. (2020) introduce an interesting audio-visual embodied task in 3D simulation environments. More recently, Dhiraj et al. (2020) collected a large sound-action-vision dataset using Tilt-bolt and demonstrates sound signals could provide valuable information for find-grained object recognition, inverse model learning, and forward dynamics model prediction. More related to us are the papers from Aytar et al. (2018) and Omidshafiei et al. (2018), which have shown that the sound signals could provide useful supervisions for imitation learning and reinforcement learning in Atari games. Concurrent to our work, Dean et al. (2020) uses novel associations of audio and visual signals as intrinsic rewards to guide RL exploration. Different from them, we use auditory event predictions as intrinsic rewards to drive RL explorations.

## 3 METHOD

In this section, we first introduce some background knowledge of reinforcement learning and intrinsic rewards. And then we elaborate on the pipeline of self-supervised exploration through auditory event predictions. The pipeline of our system is outlined in Figure 1.

### 3.1 BACKGROUND

**MDPs** We formalize the decision procedure in our context as a standard Markov Decision Process (MDP), defined as $(\mathcal{S}, \mathcal{A}, r, \mathcal{T}, \mu, \gamma)$. $\mathcal{S}$, $\mathcal{A}$ and $\mu(s) : S \rightarrow [0, 1]$ denote the sets of state, action and the distribution of initial state respectively. The transition function $\mathcal{T}(s'|s, a) : \mathcal{S} \times \mathcal{A} \times \mathcal{S} \rightarrow [0, 1]$ defines the transition probability to next-step state $s'$ if the agent takes action $a$ at current state $s$. The agent will receive a reward $r$ after taking an action $a$ according to the reward function $\mathcal{R}(s, a)$, discounted by $\gamma \in (0, 1)$. The goal of training reinforcement learning is to learn an optimal policy $\pi^*$ that can maximize the expected rewards under the discount factor $\gamma$ as

$$\pi^* = \arg\max_{\pi} E_{\zeta \in \pi} \left[ \sum R(s, t)\gamma_t \right] \tag{1}$$

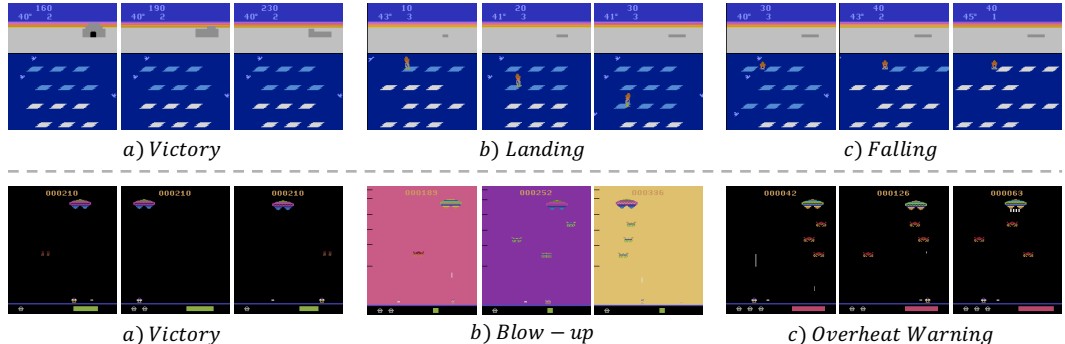

Figure 2: The first and second rows show the auditory events that we discovered by the K-means algorithm in *Frostbite* and *Assault* respectively.

where $\zeta$ represents the agent's trajectory, namely $\{(s_0, a_0), (s_1, a_1), \cdots\}$. The agent chooses an action $a$ from a policy $\pi(a|s) : \mathcal{S} \times \mathcal{A} \rightarrow [0, 1]$ that specifies the probability of taking action $a \in \mathcal{A}$ under state $s$. In this paper, we concentrate on the MDPs whose states are raw image-based observations as well as audio clips, actions are discrete, and $\mathcal{T}$ is provided by the game engine.

**Intrinsic Rewards for Exploration.** Designing intrinsic rewards for exploration has been widely used to resolve the sparse reward issues in the deep RL communities. One effective approach is to use the errors of a predictive model as exploration bonuses (Pathak et al., 2017; Haber et al., 2018; Burda et al., 2018a). The intrinsic rewards will encourage the agent to explore those states with less familiarity. We aim to train a policy that can maximize the errors of auditory event predictions.

### 3.2 REPRESENTATIONS OF AUDITORY EVENTS

Consider an agent that sees a visual observation $s_{v,t}$, takes an action $a_t$ and transits to the next state with visual observation $s_{v,t+1}$ and sound effect $s_{s,t+1}$. The main objective of our intrinsic module is to predict auditory events of the next state, given feature representations of the current visual observation $s_{v,t}$ and the agent's action $a_t$. We hypothesize that the agents, through this process, could learn the underlying causal structure of the physical world and use that to make predictions about what will happen next, and as well as plan actions to achieve their goals.

To better capture the statistic of the raw auditory data, we extract sound textures (McDermott & Simoncelli, 2011) $\Phi(s_{s,t})$ to represent each audio clip $s_{s,t}$. There are many possible ways to compute audio features. We choose sound texture mainly for its computation efficiency and effectiveness in audio-visual learning (Owens et al., 2016a;b). For the task of auditory event predictions, perhaps the most straightforward option is to directly regress the sound features $\Phi(s_{s,t+1})$ given the feature embeddings of the image observation $s_{v,t}$ and agent's actions $a_t$. Nevertheless, we find that not very effective. We hypothesize that the main reasons are: 1) the mean squared error (MSE) loss used for regression is satisfied with "blurry" predictions. This might not capture the full distribution over possible sounds and a categorical distribution over clusters; 2) the MSE loss does not accurately reflect how well an agent understands these auditory events. Therefore, we choose instead to define explicit auditory events categories and formulate this auditory event prediction problem as a classification task, similar to (Owens et al., 2016b).

### 3.3 AUDITORY EVENTS PREDICTION BASED INTRINSIC REWARD

Our AEP framework consists of two stages: sound clustering and auditory event prediction. We need to collect a small set of diverse auditory data in the first stage and use them to define the underlying auditory event classes. To achieve this goal, we first train an RL policy that rewards the agents based on sound novelty. And then, we run a K-means algorithm to group these data into a fixed number of auditory events. In the second phase, we train a forward dynamics network that takes input as the embedding of visual observation and action and predicts which auditory event will happen next. The prediction error is then utilized as an intrinsic reward to encourage the agent to explore those auditory events with more uncertainty, thus improving its ability to understand the consequences of its actions. We will elaborate on the details of these two phases below.

**Sound clustering.** The agents start to collect audio data by interacting with the environment. The goal of this phase is to gather diverse data that could be used to define auditory events. For this purpose, we train an RL policy by maximizing the occurrences of novel sound effects. In particular, we design an online clustering-based intrinsic motivation module to guide explorations. Assuming we have a series of sound embeddings $\Phi(s_{s,t}), t \in 1, 2, ..., T$ and temporarily grouped into $K$ clusters. Given a new coming sound embedding $\Phi(s_{s,t+1})$, we compute its distance to the closest cluster centers and use that as an exploration bonus. Formally,

$$r_t = \min_i ||\Phi(s_{s,t+1}) - c_i||_2 \tag{2}$$

where $r_t$ denotes an intrinsic reward at time $t$, and $c_i, i \in \{1, 2, ..., K\}$ represents a cluster center. During this exploration, the number of clusters will grow, and each cluster's center will also be updated. Through this process, the agent is encouraged to collect novel auditory data that could enrich cluster diversity. After the number of the clusters is saturated, we then perform the K-Means clustering algorithm (Lloyd, 1982) on the collected data to define the auditory event classes and use the center of each cluster for the subsequent auditory event prediction task. To be noted, the number of cluster $K$ is determined automatically in our experiments. In practice, we define $K \in [5, 30]$ for clustering at each time step and use the silhouette score to automatically decide the best $K$, which is a measure of how similar a sound embedding to its own cluster compared to other clusters. When the best $K$ keeps unchanged for 3 updates, we believe it is "saturated". We visualize the corresponding visual states in two games (*Frostbite* and *Assault*) that belong to the same sound clusters, and it can be observed that each cluster always contains identical or similar auditory events (see Figure 2).

**Auditory event predictions.** Since we have already explicitly defined the auditory event categories, the prediction problem can then be easily formulated as a classification task. We label each sound texture with the index of the closest center, and then train a forward dynamics network $f(\Psi(s_{v,t}), a_t; \theta_p)$ that takes the embeddings of visual observation $\Psi(s_{v,t})$ and action $a_t$ as input to predict which auditory event cluster the incurred sound $\Phi(s_{s,t+1})$ belongs to. The forward dynamics model is trained on collected data using gradient descent to minimize the cross-entropy loss $L$ between the estimated class probabilities with the ground truth distributions $y_{t+1}$ as:

$$L = \text{Loss}(f(\Psi(s_{v,t}), a_t; \theta_p).y_{t+1}) \tag{3}$$

The prediction is expected to fail for novel associations of visual and audio data. We use the loss $L$ as intrinsic reward. It will reward the agent at that stage and encourage it to visit more such states since it is uncertain about this scenario. In practice, we do find that the agent can learn to avoid dying scenarios in the games since that gives a similar sound effect it has already encountered many times and can predict very well. By avoiding potential dangers and seeking novel events, agents can learn causal knowledge of the world for planing their actions to achieve the goal.

## 4 EXPERIMENTS

As a proof of concept, we first conduct experiments on Atari game environments (Bellemare et al., 2013) to understand our model. And then we apply our model to more realistic 3D environments, such as Habitat (Savva et al., 2019) and TDW (Gan et al., 2020a).

### 4.1 SETUP

**Atari Game Environment** Our primary goal is to investigate whether we could use auditory event prediction as intrinsic rewards to help RL exploration. For this purpose, we use the Gym Retro (Nichol et al., 2018) as a testbed to measure agents' competency quantitatively. Gym Retro consists of a diverse set of Atari games, and also supports an audio API to provide the sound effects of each state. We use 20 familiar video games that contain sound effects to compare the intrinsic reward only exploration against several previous state-of-the-art intrinsic modules. We follow the standard setting in (Pathak et al., 2017; Burda et al., 2018b), where an agent can use external rewards as an evaluation metric to quantify the performance of the exploration strategy. This is because doing well on extrinsic rewards usually requires having explored thoroughly.

**Audio-Visual Explorations on Habitat** Following (Dean et al., 2020), we use the the apartment_0 in Replica scene (Straub et al., 2019) with the Habitat simulator for experiments. This scene has

Table 1: Category results of 20 Atari games according to the dominant type of sound effects. We label the games in which our method performs the best in **bold font**.

| Dominant sound effects | Atari games |
|---|---|
| Event-driven sounds | **Amidar**, **Carnival**, **NameThisGame**, **Frostbite**, **FishingDerby**, **MsPacman** |
| Action-driven sounds | **AirRaid**, **Assault**, **Jamesbond**, **ChopperCommand**, **StarGunner**, **Tutankham**, **WizardOfWor**, **Gopher**, DemonAttack |
| Background sounds | Asteroids, Freeway, TimePilot, BattleZone, CrazyClimber |

211 unique locations as the state space, and the action space contains three discrete actions, namely *move forward, turn right and turn left.* We follow the setting from (Chen et al., 2020) by placing one fixed audio source at a fixed location. The audio engine in Habitat could simulate various sounds at arbitrary agent receiver positions based on the room's geometry, major structures, and materials, so that the agent can hear different sounds when moving. The agent is trained without extrinsic reward, and we compare the steps needed to fully explore the scene.

**Active learning with a rolling robot on TDW** We finally test our module on a 3D multi-modal physic simulation platform, built on top of TDW (Gan et al., 2020a). As shown in Figure 7 , we place a rolling robot agent (*i.e.*, red sphere) on a billiard table. The agent is required to execute actions to interact with objects of different materials and shapes. When two objects collide, the environment could generate collision sound based on the physical properties of objects. We would like to compare the agent's behaviors in this 3D world physical environment using different intrinsic rewards.

**Baselines.** We consider four state-of-the-art intrinsic motivation modules for comparisons, including Intrinsic Curiosity Module (ICM) (Pathak et al., 2017), Random Feature Networks (RFN) (Burda et al., 2018a), Random Network Distillation (RND) (Burda et al., 2018a), and Model Disagreement (DIS) (Pathak et al., 2019). We use their open-source code to reproduce the results.

**Implementation details.** For all the experiments, we choose PPO algorithm (Schulman et al., 2017) based on the PyTorch implementation to train our RL agent since it is robust and requires little hyper-parameter tuning. We use the open-source toolbox [1] to extract sound texture features. For experiments on Atari games, we use gray-scale image observations of size 84×84 and 60ms audio clip. We set the skip frame S=4 for all the experiments. We use a 4-layer CNN as the encoder of the policy network. As for the auditory prediction network, we choose a 3-layer CNN with output channel 32, 64, 64 to encode the image observation and use 2-layer MLP with 512 hidden units to predict the auditory events. For all experiments, our model use 10K interactions for stage 1 exploration, **which is also included in the beginning of each reported curve.** The intrinsic reward in both phases are clipped to [-1, 1] after normalization. We use 8, 1, and 8 parallel environments for Atari, Habitat, and TDW experiments, respectively.

## 4.2 Experiments on Atari

**Overall Results.** Figure 3 summarizes the evaluation curves of mean extrinsic reward in 20 Atari games. For each method, we experiment with three different random seeds and report the standard deviation in the shaded region. As the figure shows, our module achieves significantly better results than previous vision-only intrinsic motivation modules in 15 out of 20 games. Another interesting observation is that the earned score keeps going up, even only using intrinsic rewards.

**Result Analysis.** We would like to provide an in-depth understanding of under what circumstances our algorithm works well. The sound effects in Atari games fall into three different categories: 1) event-driven sounds which emitted when agents achieve a specific condition (*e.g.*, picking up a coin, the explosion of an aircraft, etc.); 2) action-driven sounds which emitted when agents implement a specific action (*e.g.*, shooting, jumping, etc.) and 3) background noise/music. According to the dominant sound effects in each game, we summarize the 20 Atari games in Table 1. It is also worth noticing that, the sounds in Atari games include positive (*e.g.* earning a bonus), negative (*e.g.* dying) and meaningless sound (*e.g.* background music). None of the category accounts for the majority.

---

[1] https://github.com/andrewowens/multisensory/blob/master/src/aolib/subband.py

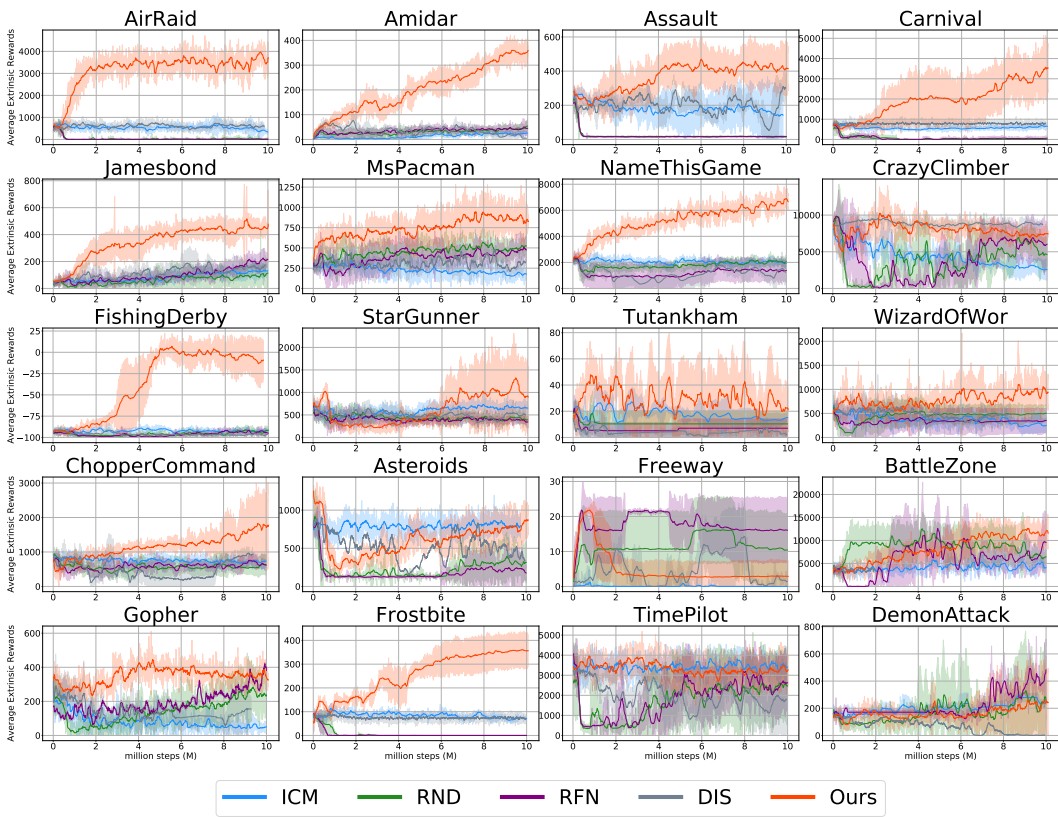

Figure 3: Average extrinsic rewards of our model against baselines in 20 Atari games. Intrinsic rewards are used for training, and the extrinsic rewards are used for evaluation only.

Based on the category defined in Table 1 and the performance shown in Figure 3, we can draw three conclusions. First, both event-driven and action-driven sounds boost the performance of our algorithm. Since the sound is more observable effects of action and events, understanding these casual effects is essential to learn a better exploration policy. Second, our algorithm performs better on the games dominant with event-driven sounds compared to those with action-driven sounds. We believe that event-driven sounds contain higher-level information, such as the explosion of an aircraft or collecting coins, which can be more beneficial for the agent to understand the physical world. Third, our algorithm falls short in comparison with baselines when the sound effects mainly consist of meaningless background noise or background music (*i.e.* CrazyClimber, MsPacman, Gopher, Tutankham, WizardOfWor, and BattleZone). These are also reasonable since our noisy agents are incentivized to make interventions that lead to making surprising sounds. Sometimes sound events will occur independently of the agent's decisions and do not differentiate between different policies, and therefore do not guide learning. The events that guide learning are those that are a consequence, directly or indirectly, of an agent's actions. The agent is encouraged to understand the causal effect of its actions and thereby take actions that lead to better exploration.

**Ablated Study.** We further carry out additional experiments to study the following questions of model design.

***Predict auditory events or sound features?*** One main contribution of our paper is to use auditory event prediction as an intrinsic reward. We conduct an ablated study by replacing this module with sound feature prediction module. We train a neural network that takes the embedding of visual state and action as input and predicts the sound textures. The comparison curves are plotted in Figure 4. We observe that the auditory event prediction module indeed earned more rewards. We speculate that auditory events provide more structured knowledge of the world, thus lead to better RL explorations.

***Sound clustering or auditory event prediction?*** We adopt a two-stage exploration strategy. A natural question is if this is necessary. We show the curve of using phase-1 only (*i.e.* sound clustering) in Figure 4. This is a novelty driven exploration strategy. The reward function is defined as the

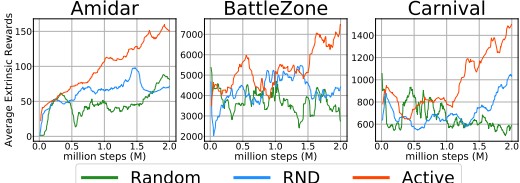

Figure 4: Comparisons on earned extrinsic rewards between our auditory event prediction module and sound feature prediction module.

Figure 5: Comparisons on earned extrinsic rewards using random, RND, and our active exploration strategy respectively.

distance to the closest clusters. We notice that the returned extrinsic reward is similar to sound feature prediction, but worse than auditory event prediction.

***Is active exploration necessary?*** We propose an online clustering-based intrinsic module for active audio data collections. To verify its efficacy, we replace this module with random explorations and RND (Burda et al., 2018a). For fair comparisons, we allow both models to use 10K interaction data to define the event classes with the same K-mean clustering algorithm. The comparison results are shown in Figure 5. We can find that the proposed active explorations indeed achieve better results. We also compute the cluster distances of these models and find that the sound clusters discovered by active exploration are more diverse, thus facilitating the agents to perform in-depth explorations.

### 4.3 EXPERIMENTS ON HABITAT

To evaluate the exploration ability of agents trained with our approach, we report the unique state coverage, given a limited exploration budget. In Figure 6, using only 16K exploration steps, our agent has already explored all unique states (211 states), while other methods have visited less than 195 unique states (190, 157, 170, and 195 for ICM, RND, RFN, and DIS, respectively) on average. Even exploring for 20k steps, all other methods have not yet fully explored the 211 states. These results demonstrate that our agent explores environments more quickly and fully, showing the potential ability of exploring the real world. We believe the reason is that our auditory event prediction encourages agents to explore the novel areas with uncertain sounds.

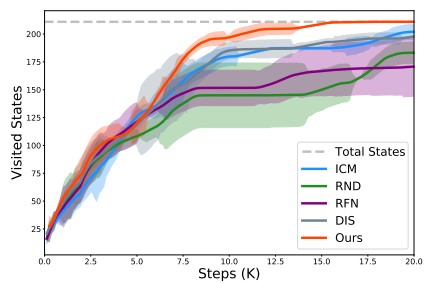

Figure 6: Comparisons of stage coverage on Habitat.

### 4.4 EXPERIMENTS ON TDW

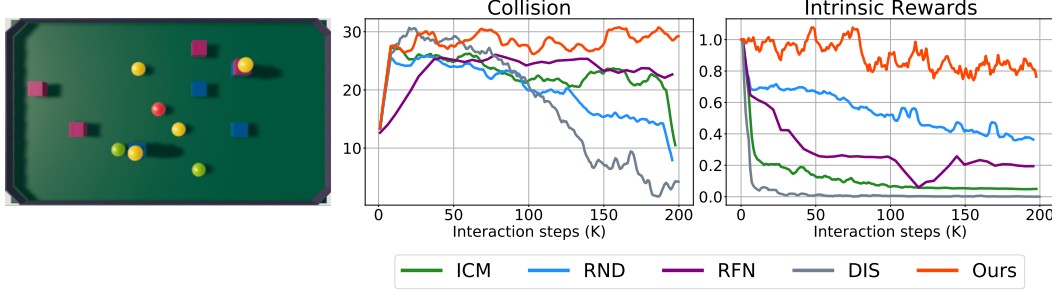

Figure 7: Explorations on a multi-modal physics environment. From left to right: physical scene, collision events, and intrinsic reward changes

**Setup.** We take an image observation of $84 \times 84$ size and 50ms audio clip as input. We use a three-layer convolutional network to encode the image and extract sound textures from the audio clip. Same as the previous experiment, we train the policy using the PPO algorithm. The action space consists of moving to eight directions and stop. An action is repeated 4 times on each frame. We run all the experiments for 200K steps with 8 parallel environments.

**Result Analysis.** To understand and quantify the agent's behaviors in this 3D physical world, we show the number of collision events and intrinsic rewards in Figure 7. We noticed two major issues with the previous vision-based curiosity models. First, the prediction errors on the latent feature space could not accurately reflect subtle object state changes in the 3D photo-realistic world, in which a physical event happens. Second, the intrinsic reward can diminish quickly during training, since the learned predictive model usually converges to a stable state representation of the environment. Instead, our auditory event prediction driven exploration will lead agents to interact more with objects in the physical world, which is critical to learning the dynamics of the environments.

### 4.5 DISCUSSIONS

This section aims to examine if our module leverages the knowledge of different auditory events for RL explorations.

***Do we need to perform event classifications?*** We aim to understand the role that the auditory event prediction module plays in our learning framework. There are two alternatives to leverage multi-sensory signals. One option is to design a simple binary classification task by predicting making a sound or not(See *Bin Cls* in Figure 8). The other one is to predict the associations between visual frames and audio clips (Dean et al., 2020) (See *SHE* in Figure 8). We will reward an agent if its prediction is incorrect. Otherwise, it will be no reward. We conduct these ablated studies in 3 Atari games, Habitat, and TDW environments. From Figure 8, we observe that predicting auditory events can indeed lead to a better exploration strategy. These results further demonstrate that understanding the causal structures of different events is essential for multi-sensory RL explorations.

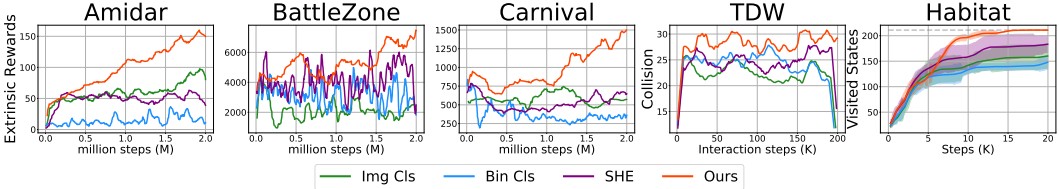

Figure 8: Ablated study on 3 Atari games, Habitat and TDW.

***Do we need audio to define events?*** We also want to understand if it is necessary to use audios to define high-level events. Perhaps, the results will also be good if we replace the audio features with visual features to cluster the event types. We plot the curves of clustering visual frame to get the event type (see *Img Cls* in Figure 8). The results are significantly worse than using audio features. We hypothesize that it might be hard to capture the patterns of pixel changes when these events happen, but sounds are more observable to discover these events.

## 5 CONCLUSIONS AND FUTURE WORK

In this work, we introduce an intrinsic reward function of predicting sound for RL exploration. Our model employs the errors of auditory event prediction as an exploration bonus, which allows RL agents to explore novel physical interactions of objects. We demonstrate our proposed methodology and compared it against a number of baselines on Atari games. Based on the experimental results above, we conclude that sound conveys rich information and is powerful for agents to build a causal model of the physical world for efficient RL explorations. We hope our work could inspire more works on using multi-modality cues for planing and control. In the future, we plan to explore more effective audio representations and combine vision-driven approaches to improve RL exploration results further.

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

## A TRAINING DETAILS

Table 2 shows the hyper-parameters used in our algorithm.

Table 2: Hyper-parameters used in our algorithms.

| Hyperparameter | Value |
|---|---|
| Rollout length | 128 |
| Number of minibatches | 4 |
| Learning rate | 2.5e-4 |
| Clip parameter | 0.1 |
| Entropy coefficient | 0.01 |
| $\lambda$ | 0.95 |
| $\gamma$ | 0.99 |

## B A NAÏVE BASELINE FOR MULTI-MODAL RL

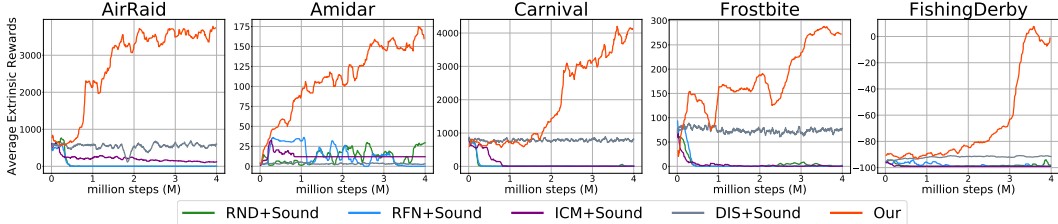

Figure 9: Average extrinsic rewards of our model against baselines combined with sound in 5 Atari games

In this section, we carry out ablated experiments to demonstrate that the gains in our method are caused by the audio-event prediction, rather than the use of multi-modality information. For four baselines (*i.e.* RND, RFN, ICM, and DIS), instead of predicting audio-event, they consider sound information by concatenating both visual and sound features to predict the image embedding in the next time step. As shown in Figure 9, our algorithm significantly outperforms other baselines in five Atari games. This indicates that it is non-trivial to exploit sound information for RL, and our algorithm benefits from the carefully designed audio-event prediction as an intrinsic reward.

## C HARD EXPLORATIONS GAMES

In this section, we investigate whether the audio-driven intrinsic rewards could be further utilized to improve policy learning in the hard exploration scenarios, where extrinsic rewards exist but very sparse. We use six hard exploration environments in Atari games, including Venture, Solaris, Private Eye, Pitfall!, Gravitar and Montezuma's Revenge for experiments. Following the strategy proposed

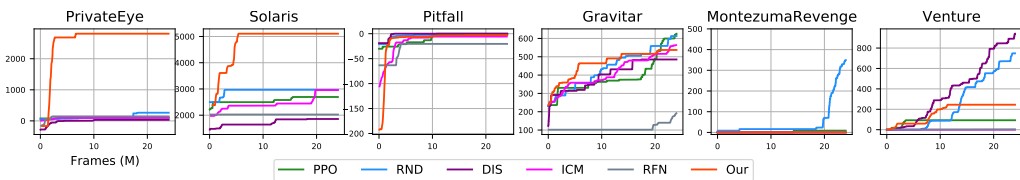

Figure 10: Comparison of combining intrinsic and extrinsic reward on 6 hard exploration Atari Games.

by RND (Burda et al., 2018a), we use two value heads separately for the intrinsic and extrinsic reward module and then combine their returns. We also normalize intrinsic rewards to make up the variances among different environments. We compare our model against those vision-only baselines including ICM, RND, RFN, DIS mentioned in the main paper. All experiments are run for 4 million steps with 32 parallel environments. We use the max episodic returns to measure the ability of explorations.

The comparison results are shown in Figure 10. Note the X-axis here refers to the frames rather than parameter updates like RND. All the baselines are reproduced using their open-source code. Our approach achieves best/comparable results in 4 out of 6 Atari games.

