# OpenReview forum: "Noisy Agents: Self-supervised Exploration by Predicting Auditory Events"
_ICLR.cc/2021/Conference — Reject_

### Official Review · AnonReviewer3 · 2020-10-26
**The paper has good idea with somewhat unconvincing experiments.**

**Rating:** 4
**Confidence:** 4

**Review:**

Summary:
This paper proposes a novel type of intrinsic rewards for RL agents based on audio events prediction. A two stage method is proposed that includes sound clustering and auditory event prediction. The auditory event prediction error is used as an intrinsic reward for better exploration of the agents. The idea is evaluated on Atari games, and two 3D simulators: Habitat and TDW.

Strengths:
1. Decent idea to use audio as curiosity to drive the exploration of an agent.
2. Well motivated by how humans integrate multisensory inputs to understand the physical world.

Weakness:
1. Justification of causal effect of its actions. See detailed comments.
2. Why not using bottom up clustering methods such as Agglomerative Clustering so that the number of clusters is not needed to be specified? The current method mannually set K to be in the range of 5 - 30, and then decide the best K is somewhat cubersome.
3. Most of the results are using Atari games as proof of concept. The results on more realistic environements are not very convincing. What are the sound events in HABITAT? If an agent just navigates in the environment listening to the same sound, how it's going to help exporation at all? There is no event defined.
4. The task is set up as an event classification task to get the intrinsic reward, so audio is actually not directly used. Technically, audio might not even be necessary in this case. What if directly using the meta data from the graphical engine to know the sound type or event type and use the ground-truth to guide the agent's learning process? This can serve as an upper bound for the proposed method to cluster audio events. Or, what if clusering the visual frames to get the event type? These are all useful baselines to demonstrate the use of audio is essential.

Detailed Comments/Questions:
1. The paper repetitively mention the agent is encouraged to understand the causal effect of its actions. However, the sound in experiments are usually just accompanying a visual event. Is it just audio-visual correspondence that helps or causality?
2. Comparison to Dean et al. 2020 should be more clear. It is mentioned this work mainly studied if the sound signals along could be utilized as intrinsic rewards, which is confusing. What is the difference compared to Dean et al. 2020 is still not clear.
3. In figure 4, how to directly use sound clustering as an intrinsic reward is not clear to this reviewer.
4. An informative baseline would be to remove clusters, but set up the task as a binary classification task: making sound or not making sound. It would be informative to know whether the agent is really leveraging the knowledge of different sound events for exploration.
5. Plenty of typos and grammar mistakes should be fixed.
    - P2, RL Explorations, A separate line of work studies adopt?
    - Sec. 3.2, to represent each audio clip s_t s_s,t ...?
    - Sec. 4.1, how well an exploration stragegy is ;  We follows -> We follow; State-of-the-arts
    ........

Justification of recommendation:
This paper proposes a nice idea, but the paper is not very well written. The experiments needs further clarifications. This reviewer is happy to raise the score if these concerns can be addressed in authors' response.




###Final Recommendation###

Based on the discussions with other reviewers and AC, this paper is not ready to publish at this stage mainy due to the following reasons:
1. the big claim of causality as also pointed out by R6
2. the writing should be significantly improved and the experiments lack details as pointed out by all reviewers
3. the new problems found during discussion with the AC regarding the ablation study, and seeds, etc.

In summary, this paper presents an interesting idea, but the experiments and writing in its current shape make the paper insufficient to be published at ICLR. The authors are encouraged to polish the paper in writing and experiments for future resubmissions.

---

> ### Author Response · Authors · 2020-11-21
> **Response to AnonReviewer3**
>
> We thank the reviewer for your extremely constructive comments to strengthen this work. Several ablated studies you suggested indeed help us to have an in-depth understanding of how and why the auditory event prediction module works.
> We will also carefully proofread the typos. Below we provide point-to-point responses.
>
> > Justification of the causal effect of its actions. Comparisons with audio-visual correspondence [1].
>
> -  We first would like to reiterate that this is a concurrent work of using multi-sensory cues to drive RL exploration.
> -  We do agree that this is an interesting baseline to understand how to best leverage the audio cues (correspondence or casual). We conduct experiments on Atari games, Habitat, and TDW. The results show that our model using auditory event prediction consistently outperforms their model in most of the environments.  These results further demonstrate effective of using the causal effect of its actions (through auditory events prediction) to drive RL exploration.
>
> > Experimental setting on Habitat.
>
> We follow the audio-visual navigation setting used [1,2]  by placing a fixed audio clip to a fixed location. The audio API  in Habitat could simulate various sound waveforms received at arbitrary agent positions based on the room’s geometry, major structures, and materials. Even though there is only one sound source, the agent hears differently in each location, Our model could then leverage the discovered latent auditory events to drive explorations.
>
> >  Alated study on binary classification on making sound or not.
>
> Thanks for your suggestion.  We implement a baseline to encourage agents to explore the state that makes the wrong prediction of making a sound or not.
>
> -  Unfortunately, it can not work in audio-visual navigation on Habitat, since most locations in the room could hear the sound.  The prediction task is trivial.
> -  The results in Atari games and TDW are not as good as predicting auditory events.
>
> These results further verify the importance of using a harder task (e.g. predicting underlying sound events) to drive RL exploration.
>
> > Alated study on using visual frames to define event classes.
>
> Thanks for your suggestion! We implement a baseline using visual data to define the event classes on Atari, Habitat, and TDW.  Specifically, we first extract the features with a pre-trained ResNet-18 from a 10K interaction visual frame and then run a clustering. And then we use the visual data alone to drive the event-prediction baed RL exploration. The experiment results show that our model could achieve much better results than this baseline. The results demonstrate the use of audio to define events is essential.
>
>
>
> > Agglomerative Clustering.
>
> This is a nice suggestion. We try this method and it indeed can automatically discover auditory events and achieve similar exploration results. The downside is that the implementation of this model is not very fast in the existing python sklearn library. In our experiment, we use a fast K-means package in VLFeat. We will mention this alternative approach in the revision.
>
> >Details of the sound clustering baseline in Fig. 4.
>
> We are sorry for the confusion. Our exploration model consists of two-stage: sound clustering and auditory event prediction.  In the first stage, we train an RL policy that rewards the agents based on sound novelty. The sound clustering baseline here means that we only use stage-1 for exploration.  This is a novelty driven exploration strategy. The reward function is defined as the distance to the closest clusters. We will make this more clear in the revision.
>
>
>
> [1] Victoria Dean, Shubham Tulsiani, Abhinav Gupta. See, Hear, Explore: Curiosity via Audio-Visual Association. https://arxiv.org/abs/1912.11474
>
> [2] Changan Chen, Unnat Jain, Carl Schissler, Sebastia Vicenc Amengual Gari, Ziad Al-Halah, Vamsi Krishna Ithapu, Philip Robinson, Kristen Grauman. SoundSpaces: Audio-Visual Navigation in 3D Environments. ECCV 2020
>
>
>
> Please don’t hesitate to let us know if there are any additional clarifications or experiments that we can offer.

---

> > ### Author Response · Authors · 2020-11-23
> > **[Revision uploaded] Look forward to your feedback!**
> >
> > Thanks again for your constructive comments. We have made substantial changes in the revision according to your review. In particular, we’ve included detailed ablation studies (section 4.5 and figure 8). As the discussion period is about to end, please don’t hesitate to let us know if there are any additional clarifications that we can offer, as we would love to convince you of the merits of the paper. Thanks!

---

### Official Review · AnonReviewer4 · 2020-10-27
**Removing the earplugs from RL**

**Rating:** 5
**Confidence:** 3

**Review:**

**Update**: Other reviewers have pointed out issues with this paper's ablation study. Additionally, it is difficult to trust the empirical results because they are based on only three runs. In light of these criticisms, I have updated my score from a 7 to a 5. I still think this idea is neat and am generally a proponent of introducing audio into work on RL, but the experiments as presented in this submission do not currently paint a complete picture.

**Summary**: This paper investigates sound perception as a means of intrinsic reward for reinforcement learning agents. Specifically, the proposed method rewards the agent for discovering state/action pairs which lead to sound events that are difficult to predict. Especially in environments where sound effects are correlated with high-level events, this strategy tends to improve performance over sound-agnostic baselines.

Overall, I think this is an interesting paper. Audio events provide important queues which help humans understand the natural world and influence their decision. Such rich acoustic structure is often imitated in simulated environments, even in early Atari games. While substantial effort has gone into extracting as much usable information as possible out of the pixels of Atari games, the easily-accessible sound systems in these environments have been mostly ignored. It stands to reason that, especially for particular games, this information could be helpful.

Hypothesizing that encountering novel audio events may be informative for learning good policies, the authors propose a strategy which explicitly rewards agents for finding such states. They show that this strategy leads to improved performance on several environments, especially ones where audio information is associated with high-level events and interactions (e.g. balls colliding) as opposed to background noise (e.g. music in games).

From an audio perspective (my primary area of expertise), the proposed strategy seems reasonable. The authors cluster perceptually-informed embeddings of audio slices to identify discrete classes, which makes sense when targeting environments whose audio primarily consists of sound effects (e.g. a clacking sound) synchronized to high-level events (e.g. two balls colliding). Why did the authors choose to use a texture-based sound embedding (McDermott & Simoncelli, 2011) as opposed to something more standard like log-amplitude Mel spectrograms? I would be quite interested to see how the performance of the latter compares to texture-based embeddings.

The experiments also seem reasonable: they compare the performance of the same policy-learning model/algorithm with different intrinsic reward sources on a slew of sparse-reward environments. The results seem to indicate that the proposed intrinsic reward usually leads to the best policy among the examined sources of intrinsic reward. One criticism is that it would be nice to see if these different intrinsic reward strategies are symbiotic; i.e., if multiple sources can be combined to improve performance. It would also have provided stronger evidence in favor of the proposed intrinsic reward function to see results on different policy-learning algorithms (besides PPO and CNNs). A caveat here is that I have limited experience on RL and defer to the expertise of the other reviewers in assessing the relevance of the baselines (ICM, RND, RFN, DIS).

The authors need to do more to distinguish their work from that of Omidshafiei et al. 2018. In a cursory overview of that work, it appears that there is a lot of overlap with the proposed work. It seems that the primary distinction is that the proposed work uses audio _only as part of the intrinsic reward_ (leaving the policy model unaware of audio cues), while the prior work _adds audio processing to the agent_. Can the authors comment further on the distinctions between their work and this prior work? Also, it's not immediately clear from the paper if the policy model (agent) receives audio as part of its state input; can the authors clarify?

---

> ### Author Response · Authors · 2020-11-21
> **Response to  AnonReviewer4**
>
> We thank the detailed and constructive comments from the reviewer.
>
> > Why choose sound textures?
>
> Thanks for your question. Our framework is very general. We could also use other pre-trained deep learning-based feature extractors on Mel spectrogram to represent audio clips.   We choose sound textures in the experiment for two reasons.
>
> - Extracting sound texture is more computationally efficient compared with other deep learning-based methods.
> - Sound textures have been successfully used in prior work on audio-visual learning [1,2]. We use their implementations for the new audio-visual RL exploration application.
>
> > Distinctions between our work and [3].
> -  We would like to clarify that the primary goal of our work is to design an audio-driven intrinsic module for RL exploration without extrinsic reward.
>  - In [3], they study how to use multi-sensory data to improve RL policy learning when extrinsic rewards available, which is a different research problem.
> - We adapt this baseline to our intrinsic reward only setting in section B in supplementary materials. Concretely,  we concatenate both visual and sound features as input.  We use the predictions on image feature embeddings instead of the auditory events as reward functions for the intrinsic module. The results are significantly worse than our newly designed AEP module, which demonstrates the importance of developing a practical audio-driven intrinsic reward function to drive RL explorations.
>
>
> [1] Andrew Owens, Jiajun Wu, Josh McDermott, William T. Freeman, Antonio Torralba. Ambient Sound Provides Supervision for Visual Learning. ECCV 2016
>
> [2] Andrew Owens, Phillip Isola, Josh McDermott, Antonio Torralba, Edward H. Adelson, William T. Freeman. Visually Indicated Sounds. CVPR 2016
>
> [3] Shayegan Omidshafiei, Dong-Ki Kim, Jason Pazis, Jonathan P. How. Crossmodal Attentive Skill Learner. AAMAS 2018.
>
> Please let me know if you have any other questions.

---

### Official Review · AnonReviewer1 · 2020-10-28
**Official Blind Review #1**

**Rating:** 6
**Confidence:** 4

**Review:**

Summary:
This work proposes the use of sound prediction as intrinsic reward to guide reinforcement learning (RL) exploration. Sounds are often directly related to causal effects of actions and interactions. By modeling different sound event classes via a clustering algorithm, the cluster prediction errors are used as intrinsic rewards for RL. The proposed approach is tested on three different simulation environments. The results on 22 different environments (20 Atari games, Habitat and TDW environments) illustrate that using exploration capability of the proposed approach and improved performance in comparison to other baseline methods leveraging intrinsic reward modules for RL exploration.

########################

Pros:
- The idea is novel and of interest to the RL community at large. The approach is clear and easy to follow.
- Comprehensive experimental analysis and convincing results. Specifically, the analysis of dominant sound types in Atari games is well done. It helps understand that the Atari games for which there are no gains with the proposed method have dominantly background sounds. It also helps isolate the games in which sounds are causal effects of actions or events.
- The ablative analysis demonstrates well that sound event prediction is a beneficial way to use audio as intrinsic rewards.
- Testing on different variety of test beds also highlights the exploration capability of the proposed work in comparison to other intrinsic reward methods.

########################

Cons:
- The design choice of using texture features for audio is unclear and not motivated well. What properties do they capture that are desirable for the RL domains used for experimentation? It is also unclear how these are computed. Some more insights about the choice of audio features will be helpful to the reader.
- If distances between sound texture features do not capture causal structure of the auditory events, then why are they suitable to be used for clustering? Wouldn’t the same problem carry over? Why should clusters be assumed to capture inherent causal structure when distances between sound texture features are used to create the clusters in the first place?
- The writing requires more clarity (suggestions below).

########################

Reason for score:
The idea is novel and results presented look great. The writing needs some rework (suggestions for improvements below) along with some explanations for feature computation and experimental conditions. I recommend for acceptance provided the authors revise the manuscript as per requested changes.
---
UPDATE:
Due to concerns raised by other reviewers, and my own confusion about the computation of audio features, and clarifications on the causality stance, I have lowered my score from 8 to 7.

########################

Questions during rebuttal:
- Please refer to the questions in the Cons section and other feedback.
- Comparisons are performed with methods using intrinsic motivation modules. How would the proposed work compare with other prior work where audio is used to aid RL (not necessarily as an intrinsic module), such as Aytar et al., 2018; Omidshafiei et al., 2018?


########################

Some typos and other feedback:
- Consider using the \citet command when referring to the authors of a reference paper in a sentence. (Eg: “Silver et al., 2016 concluded that…” versus “(Silver et al., 2016) concluded that …”)
- Introduction, paragraph 1: “…a range of intrinsic reward function.” -> “…a range of intrinsic reward functions.”
- Introduction, last paragraph (prior to bullets): provide citations for the Atari domain, and expand the first usage of TDW.
- Related Work, RL explorations: Tompson sampling -> Thompson sampling
- Related Work, RL explorations: Osband reference missing the year in the references section and the citation.
- Related Work, RL explorations: “…uses the errors of predicting the next state…” -> “use the errors of predicting the next state…”
- Figure 1 caption: “The agent start to collect a diverse set…” -> “The agent starts to collect a diverse set..”
- Figure 1 caption: “..and then cluster them into …” - > “..and then clusters them into …”
- Figure 1 caption: “.. the agent use errors of auditory events…” -> “.. the agent uses errors of auditory events…”
- Figure 2 caption: end the sentence with “respectively”
- In several places: Forward dynamic network -> forward dynamics network
- Section 3.2: space missing between “s_v,t” & “and”
- Section 3.2: “..agent’ actions A_t” -> “..agent’s actions a_t”
- Section 3.2, paragraph 2, sentence 1: The audio clip is represented by s_t or s_s,t?
- Section 3.3, paragraph 1, last sentence: “..details of these two-phase below” -> “..details of these two phases below”
- Section 3.3, Sound clustering: Formally define silhouette score or give an insight for what it captures for the reader.  What is the criteria to determine a new cluster will be created?
- Section 3.3, Sound clustering: “automatic decide the best K” -> “automatically decide the best K”
- Section 3.3, Auditory event predictions, last paragraph: “It will reward the agent at the that stage…” This is a grammatically incorrect sentence. Possible correction could be: “It will reward the agent at that stage and encourage it to visit more such states since it is uncertain about this scenario.”
- Section 3.3, Auditory event predictions, last paragraph: “…avoid dying scenario in the game…” -> “…avoid dying scenarios in the game..”
- Section 3.3, Auditory event predictions, last paragraph: “.. and keeping seeking novel events…” -> “.. and seeking novel events…”
- Section 4, paragraph 1: Provide citations for Atari, Habitat and TDW environments.
- Table 1 caption: “..bold front” -> “..bold font”
- Section 4.1: Consider formally defining “extrinsic rewards” since it is used several times in the draft.
- Section 4.1, Atari Game Environment: “..also support an audio API..” -> “..also supports an audio API..”
- Section 4.1, Atari Game Environment: “..contain the sound effects to compare..” -> “..contain sound effects to compare..”
- Section 4.1, Atari Game Environment: The last sentence is grammatically incorrect. Consider replacing it with “We follow the standard setting in Pathak et al, 2017; Burda et al., 2018b, where an agent can use external rewards as an evaluation metric to quantify the performance of the exploration strategy. This is because doing well on extrinsic rewards usually requires having explored thoroughly.”
- Section 4.1, Audio-Visual Explorations on Habitat: “..with Habitat simulator for experiments.” ->  “..with the Habitat simulator for experiments.”
- Section 4.1, Audio-Visual Explorations on Habitat: “We follows the setting from …” -> “We follow the setting from …”
- Section 4.1, Audio-Visual Explorations on Habitat: “… can hear the different sound when moving.” -> “…can hear the different sounds when moving.”
- Section 4.1, Baselines: four baselines are used instead of five as stated in this paragraph.
- Section 4.1, Baselines: “state-of-the-arts” -> “state-of-the-art”
- Section 4.1, Implementation details: “..our model use 10K interaction for stage 1…” -> “..our models use 10K interactions for stage 1…”
- Section 4.2, Result Analysis: What is implied by positive, negative and meaningless sounds? Examples?
- Section 4.2, Result Analysis: “.. our algorithm works well under what circumstances.” -> “.. under what circumstances our algorithm works well.”
- Section 4.2, Result Analysis, paragraph 2: “.. event-driven sounds compare with those with action-driven sounds” -> “.. event-driven sounds compared to those with action-driven sounds”
- Section 4.2, Result Analysis, paragraph 2: “..when the sounds effects mainly consist…” -> “..when the sound effects mainly consist…”
- Section 4.2, Sound clustering or auditory event prediction and Fig 4.: It is unclear how the Clustering only condition is different from the proposed approach? What is the loss for clustering only that is used as intrinsic reward?
- Section 4.3, last sentence: Several grammatical errors. Consider replacing with: “We also compute the cluster distances of both models and find that the sound clusters discovered by active exploration are more diverse, thus facilitating the agents to perform in-depth explorations.”
- Section 4.3, sentence 1: Consider replacing with: “To evaluate the exploration ability of agents trained with our approach, we report the unique state coverage, given a limited exploration budget.
- Section 4.4, Setup: “The action is repeated 4 times on each frame.” -> “An action is repeated 4 times on each frame”
- Section 4.4, Result Analysis: “.. intrinsic reward rewards in Figure 7.” -> “..intrinsic rewards in Figure 7.”
- Section 4.4, Result Analysis: “..prediction errors on latent features space…” -> “..prediction errors on the latent feature space..”
- Conclusion: “… prediction as an exploration bonuses, which allows RL agent…” -> “… prediction as an exploration bonus, which allows an RL agent…”
- Conclusion: “Based on the experimental result above, we therefor conclude that sound conveys…” -> “”Based on the experimental results above, we conclude that sound conveys…”

---

> ### Author Response · Authors · 2020-11-22
> **Response to AnonReviewer1**
>
> We thank the reviewers for your detailed comments to strengthen this work! Below we provide a point-to-point response.
>
> >Motivations of using sound texture extraction.
>
> Please first refer to the common concerns to see the recent progress on audio-visual learning using sound texture features.
>
> - We agree that our framework is general, and there are many possible ways to compute audio features.
>
> - We choose the perceptually inspired sound texture mainly for its efficiency and effectiveness on audio-visual learning.
>
> > Details of sound texture extraction.
>
> - We first apply normalization to the audio waveforms to [-1, 1] and filter them with a bank of 32 bandpasses.
>
> - Next, we take the Hilbert envelope of each channel and resample them to 400 Hz.
>
> - We then calculate the statistics of these envelopes, including the Pearson correlation between pairs of channels, the mean and the standard deviation of each frequency channel,  and the modulation power.
>
> - Finally, we concatenate them as the sound texture.
>
> In our experiment, we represent each audio clip is a 533-dimensional  vector.
>
>
>
> > Clarifications on the causality instance.
>
>  Sorry for the confusion.  We summarize the following claims and hope to resolve your confusion. We will also revise the paper and make it more clear, Thanks for pointing this out!
>
> - The distances between sound texture features can capture the causal structure of the auditory events.
>
>  - The sound texture features are also suitable to be used for clustering events.
>
> - The main point we want to make is that directly regressing sound textures as an intrinsic reward is not optimal to capture the causal structure of auditory events.
>
> - We hypothesize that there might be two possibilities:  1) the MSE loss (L2 distances) used for regressing the sound textures is bad with uncertainty between event clusters, but the cross-entropy loss using for predicting event clusters solves this problem well;   2) MSE is satisfied with "blurry" predictions, and might not capture the full distribution over possible sounds as well as a categorical distribution over clusters.
>
> - We will add two new baselines to verify the importance of predicting the auditory events, as an intrinsic reward, including binary classification of making a sound or not, and audio-visual correspondence.
>
>
>
> Thanks for your careful proofreading again and we will revise the draft accordingly. Please also let us know if there are any additional experiments and clarifications that we could provide.

---

> > ### Comment · AnonReviewer1 · 2020-11-24
> > **Follow up by R1**
> >
> > Thanks for your response to my questions and for the additional results you have added. I have a few lingering questions and it would be beneficial to hear more about them from the authors:
> >
> > - Could you give more details about what do sound texture features convey? In the draft, you mention the example of sound textures capturing a critical event like an explosion versus the intricate rhythm of the bang. What is it about sound textures that allows it to make such a distinction? How might they be different from MFCC or SFTF features? Why are they computationally efficient compared to other pretrained deep learning based features (such as VGGish features https://github.com/tensorflow/models/tree/master/research/audioset/vggish)?
> >
> > - A follow up to your discussion with R3 about causality versus correlation: In general I understand your position, but is it possible that some sound events are correlated to the agent's action but not causal of the agent's action? Is causality guaranteed? Moreover, are novel sound events the best way to explore more? Good exploration strategies in some states of the environment might not produce any sounds.  Could your method augment or be combined with other exploration methods?
> >
> > I would reiterate that the authors add the discussion points and additional feedback from all reviewers in the final version of the paper.

---

> > > ### Author Response · Authors · 2020-11-24
> > > **Follow-up Response to Reviewer1**
> > >
> > > Dear reviewer 1:
> > >
> > > Thanks for your question!
> > >
> > > >  Sound textures or VGGish features.
> > >
> > > - Sorry for the confusion. we have removed the claim of sound textures capturing a critical event rather than the rhythm of the bang. We have also rewritten our explanations on why directly predicting feature embedding not optimal in section 3.2.
> > >
> > > -  We actually start with VGG-like pre-trained features on the sound spectrogram and get similar good or even slightly better exploration results,  but there might be 2-3 times slower.   The reason is that we need first to convert the raw waveform to a spectrogram and then do a forward pass using VGG.
> > >
> > > - We would like to reiterate that using sound texture features is not our main contribution.  We also don't observe that sound textures can make a big distinction. We agree that designing effective audio representations for RL exploration is a very exciting direction to go. We have pointed out that in the revision (See Sec. 5).
> > >
> > > > Combined with other exploration results.
> > >
> > > Thanks for this great idea.  We indeed can combine the intrinsic rewards of our audio-driven with another vision-driven approach (e.g., RND) to cover more states of the environment. This strategy might further improve RL exploration results. We have also pointed out that in the revision (See Sec. 5).
> > >
> > >
> > > > Paper revision.
> > >
> > > Thanks again for your carefully proofreading. We have already incorporated them into the revision draft. Please let us know if you have additional comments on the updated draft!

---

### Official Review · AnonReviewer2 · 2020-10-30
**Multisensory approach to explore multisensory environments with some ambiguities**

**Rating:** 4
**Confidence:** 3

**Review:**

The manuscript proposes a novel method on how to include audio signals as rewards for image-based reinforcement learning (RL) exploration. More specifically, the agent collects audio signals, identifies the individual audio sources and trains a classifier to forecast the audio cues. The forecasting error in the next time step is used as reward for RL. This encourages the behaviour to explore novel audio cues. Authors’ apply the method to different set of problems (Atari games, Habitat simulator and active learning TDW simulator) and find that the proposed method improves the agent’s behaviour/performance.

Combining audio and video information is essential to improve the explorative behaviour of agents that explore multisensory environments, especially if the relationships between agent behaviour and sounds are causal. In this case, the audio signals provide useful information much earlier in comparison to time than images could. Hence, the proposed approach is relevant for improving the explorative behaviour of agents in multisensory environments.

The paper is well structured, and the description of the methods is clear. The description of the experiments is not so clear. Note, that the reviewer is not familiar with these environments. For example, in the Habitat experiment, how many sound sources have been used? One in each room? If additional sources were strategically placed in different rooms to favour the proposed method, then one can expect that performance increases. Also, the reviewer found it challenging to interpret the results. For example, Figures 3 shows that the reward of the proposed methods is higher than the reward of other methods. It is not clear, however, how much the performance of the different methods is better than chance level performance. High reward means better performance, however, was the agent able to win game levels? Would be helpful to learn more about how “good” the agent was able to perform the task. Or to clarify which reward level was needed for the agent to successfully explore the environment. Moreover, when assessing the random exploration performance, how was random exploration performed? Each action was randomly selected based on a uniform distribution. If so, then this may not be the most reliable way to estimate random exploration.

Overall, the paper proposes an interesting approach that enable agents to autonomously explore an environment based on unknown auditory cues.

The pros:
-	Integration of multisensory information in multisensory environments is essential when the information provided by the different sensors is independent and hence contributes to development of more successful behaviour.
-	The idea to use the prediction error to encourage explorative behaviour is smart
-	Method evaluated in on different environments

The cons:
-	Interpretation of performance is not straight forward
-	A more systematic analysis of the relationship between relevant sound sources and environment sounds would be helpful to get an idea about the potential and constraints of the proposed method


### Update after discussion period ###
Good idea, but the results don't clearly support the authors claims. I lowered my score.

---

> ### Author Response · Authors · 2020-11-21
> **Response to AnonReviewer2**
>
> We would like to thank the reviewer for your instructive comments to improve the draft.
>
> > Experimental setting in Habitat.
>
> We are sorry for the confusion.  We will clarify the experimental setting on Habitat in the general response.
>  There is only one sound source in this apartment, but the agent hears it differently in each location.
>
> > Interpretation of performances.
> -  Using extrinsic rewards to evaluate intrinsic only motivated agents is a standard-setting in [1, 2]. Doing well on extrinsic rewards usually requires having explored thoroughly.
> - The results on Atari are more like a proof-of-concept. Our experiments on Atari do not aim to evaluate if the agent can win the game using an intrinsic reward. Since Atari games contain a variety of audio environments,  we hope to have an in-depth study of our module's potential and limitations (see Table 1).
>
>
> -> The random exploration baseline in Figure 5.
>
>  > Thanks for your suggestion. We will add a baseline by using RND to collect 10K interaction data. We will update these results in the revision.
>
> Please let us know if there are any additional clarifications and experiments that we can provide.

---

> > ### Comment · AnonReviewer2 · 2020-11-23
> > **Follow up**
> >
> > Thank you for the clarifications and for adding random basline experiments in Figure 5. No more open questions.

---

> > > ### Author Response · Authors · 2020-11-23
> > > **Thanks for your comments!**
> > >
> > > Dear reviewer2:
> > >
> > > Thank you so much for our very constructive comments, which have helped us strengthen this work significantly!

---

### Official Review · AnonReviewer5 · 2020-11-10
**Interesting idea. Some implementation details are missing. Intuition is missing.**

**Rating:** 5
**Confidence:** 4

**Review:**

The paper introduces an approach for using auditory event prediction to drive exploration, specifically by using the prediction error of auditory events as a shaping reward which is used along with the extrinsic reward. The approach consists of two phases. In the first phase, the agent explores the environment in a self-supervised manner and collects audio data, which are then clustered. The index of these clusters are the auditory events used during the second phase. In the second phase, an RL agent learns to predict the label of an auditory event at the next time step from the current observation and an action. The prediction error produced from this classification task is given as a shaping reward along with the extrinsic reward to the learning agent. The demonstrated results seem to suggest that this approach can improve over other self-supervised exploration methods in Deep RL.

Pros:
Overall, the approach of using auditory events to drive exploration is novel.
The paper is well-written and presents the idea clearly.

Cons:
Some implementation details about the approach are not available.
The approach seems to require a pre-training phase that requires collecting a significant amount of data.
Does not provide a clear intuition for why auditory events lead to better performance than other self-supervised approaches.

Questions:
I have a few questions related to the approach, experiment results, and implementation details, which would help clarify my understanding of the paper.
1. How are the intrinsic and extrinsic rewards combined to train an RL agent? The paper doesn’t seem to have any details about this. I assume it is the sum of extrinsic reward and a scaled-down intrinsic reward? If yes, then how was the scaling factor for the intrinsic reward chosen? And how was this chosen for the baseline methods?
2. The approach seems to require a pre-training phase to identify different auditory events through K-means clustering. And this pre-training phase relies on generating behavior to seek out novel clusters. Would it be possible to avoid such a pre-training phase, and instead identify the auditory events as the agent learns to solve the main task? Would be interesting to see if the performance still holds in this scenario.
3. In the results, the amount of data that was used for pre-training doesn’t seem to be included. It seems unfair to the other methods when the introduced method has experienced more data because of the pre-training phase. A fairer comparison would be to train the baseline agents longer to account for the pre-training phase and then make a comparison with the introduced method?
4. In many of the Atari games, the baseline approaches seem to fail in learning? Could the authors comment on why this is the case?
5. The main experiments (Fig. 3) on Atari seem to use 8 parallel environments. The choice of the number of environments would affect the batch size of the learning update and usually, it is common to use 16 or 32. Why was this choice made?

---

> ### Author Response · Authors · 2020-11-19
> **Response to AnonReviewer5**
>
> Dear Reviewer5, thank you very much for the detailed review! Before we address your specific questions on the Atari, we would like to mention that we have other main experiments on Habitat and TDW as well. Experiments on Atari is mainly used for understanding our model.
>
> >How to combine intrinsic results and extrinsic results?
>
> Thanks for your question. But the reviewer might misunderstand our experimental setting.
> - The main results reported in the paper  (Fig 3, 4, 5, 6, and 7) all use intrinsic rewards only for exploration. The extrinsic rewards are used for the evaluation, not for training.
> - Using extrinsic rewards to evaluate intrinsic only motivated agents is also a standard-setting for in [1, 2]. It serves as a proxy for assessing whether or not an agent explored the environment well. Doing well on extrinsic rewards usually requires having explored thoroughly. It is also practical in many robotic and embodied AI applications, which require to actively interact with the world without extrinsic rewards.
>
> > Pre-training data is not included. Unfair comparisons with baselines.
>
> Thanks for your concern. But the reviewer might misinterpret the results.
> - We have clearly clarified this fact in the implementation details section.
>     "For all experiments, our model uses 10K interaction for stage 1 exploration, which is also included in the beginning of each reported curve."
> - The main experimental curves of Fig 3, 6, and 7 have included these frames in the beginning. So the comparisons with the baselines are fair.
>
> > Identify the auditory events as the agent learns to solve the main task.
>
> Thanks for your suggestions. We are not sure if we understand your question correctly. It will be nice if you could elaborate on more details.
> - We have run an ablated study in Figure 4 (clustering only). This model only uses stage-1 for explorations by just finding the novel sound. It can still learn something, but not as good as our two-stage method.
> - We use a softmax loss function for our event-prediction-based curiosity model. The softmax function requires a pre-defined number of classes. Changing the number of classes during model training will make it unstable.
>
> > Collecting a significant amount of data for pre-pre-training data collection.
>
> Thanks for your question.
> - We only use 10K data for stage-1 exploration. It is quite small compared to the full data  (e.g. 10 million on Atari) using for policy training.
> - We also try different numbers of actions for stage-1 exploration. For Frostbite, the maximum extrinsic rewards for 5k, 10K, and 30K are 176, 185, 179. For NameThisGame, the rewards are 3506, 4689, and 4653. 10K actions for data collection are enough to get good scores
>
> > Implementations on Atari
>
> Thanks for your questions.
>  - We use their open-source codes to reproduce the baselines, and the results are consistent with the [1,2].
>  - We do find that there are some Atari environments that visual-only methods fail to learn.   We speculate that the vision-only model fails to capture subtle differences from pixels in these environments. But sound instead gives a more discriminative signal to guide explorations.
>  -We use 8 parallel environments on Atari mainly for the computation resource constraints. Since all the models use the same experimental setting, we believe the comparisons are fair.
>
> [1] Yuri Burda, Harri Edwards, Deepak Pathak, Amos Storkey, Trevor Darrell, and Alexei A Efros. Large-scale
> 325 study of curiosity-driven learning. ICLR, 2018.
>
> [2] Deepak Pathak, Pulkit Agrawal, Alexei A Efros, and Trevor Darrell. Curiosity-driven exploration by
> 385 self-supervised prediction. In ICML, 2017.
>
> Please let us know if you have any other questions!

---

> > ### Author Response · Authors · 2020-11-23
> > **Look forward to your feedback!**
> >
> > Dear Reviewer 5,
> >
> > Thanks again for your constructive review, which has helped us improved the quality and clarity of the paper. In addition to our response above, in the revision, we have included comparisons with additional baselines.
> >
> > As the discussion period is about to end, please don’t hesitate to let us know if there are any additional clarifications that we can offer, as we would love to convince you of the merits of the paper. We appreciate your suggestions. Thanks!

---

> > > ### Comment · AnonReviewer5 · 2020-11-23
> > > **Follow-up Comment**
> > >
> > > Thanks for clarifying the misunderstandings I had about the paper!
> > >
> > > I would recommend the authors to clarify the captions around Fig. 3 stating that the intrinsic rewards are used for training and that the extrinsic rewards are used for evaluation only.
> > >
> > > After reading the other reviews, authors' responses, and the planned changes, I think that this paper is a good contribution that explores combining auditory events with visual Deep RL agents. I am increasing my score to 6.

---

> > > > ### Author Response · Authors · 2020-11-23
> > > > **Thanks for your comments!**
> > > >
> > > > Dear reviewer5:
> > > >
> > > > That’s great to hear. We’d like to thank you again for your very constructive comments, which have helped us improve the quality of the paper significantly. We will update Fig. 3 in the revision.

---

### Official Review · AnonReviewer6 · 2020-11-29
**Promising but not ready**

**Rating:** 2
**Confidence:** 4

**Review:**

# Summary

This paper investigates the incorporation auditory events into reinforcement learning. Specifically, it proposes a new algorithm that uses event prediction as an intrinsic reward. This algorithm has two phases. In the first phase, an agent is given a small number of episodes to gather diverse auditory data. This phase has two pieces that work in conjunction: 1) As it learns, the agent clusters the sound embeddings it has observed using K-means. 2) To encourage the agent to find diverse auditory data, it is rewarded for reaching states that emit sounds that are far away from its existing cluster centers. In the second  phase (which dominates the first in terms of number of episodes consumed), a new agent is trained to predict the cluster center to which the next auditory event will belong. It is rewarded according to how incorrect its prediction is, thereby encouraging the agent to explore states for which it has difficulty predicting the auditory event.

The paper performs experiments in Atari, Habitat and TDW. It compares against strong vision-based intrinsic reward baselines. It also performs experiments comparing its proposed methodology to ablations with the aim of determining 1) whether it suffices to predict sound features instead of auditory events, 2) whether a two-stage exploration strategy is necessary, 3) whether it is necessary to perform active exploration in the first phase, and 4) whether event classification is necessary.

# Writing

This submission does not read as that of a paper ready for publication. Its organization, unnecessary use of the passive voice, singular-plural inconsistencies, tense mixing, and muddled descriptions weaken its value. Below is a (non-exhaustive!) list of issues.

- Abstract

“We first conduct an in-depth”

There needs to be a transition from the method description for this “first” to fit here.

- Introduction

Why is Deep Reinforcement Learning all caps?

“algorithms aim to learn a policy of an agent to maximize its cumulative rewards by interacting with environments”

singular/plural

“domains, such as video game”

Game should be plural

“While these results are remarkable, one of the critical constraints is the prerequisite of carefully engineered dense reward signals, which are not always accessible.”

Is Go a good example of this? AlphaZero accomplishes the same task without carefully engineered dense reward signals.

“For example, curiosity-driven intrinsic reward based on prediction error of current (Burda et al., 2018b) or future state (Pathak et al., 2017) on the latent feature spaces have shown promising results.”

This sentence is ordered awkwardly. As written, “have” refers to “curiosity-driven intrinsic reward”, which is singular. “on the latent feature spaces” doesn’t work here.

“visual state is high-dimensional” -> states are

“speech or other nonverbal but audible signals” -> speech or other audible signals OR speech or nonverbal, audible signals

Using “other” here makes it read as if speech is a member of “nonverbal but audible signals”

“However, it is just as much in physics.”

 ?

“A naive strategy would be” -> A naive strategy is

“In the beginning”

Does this mean in the first phase?

“The state that has the wrong prediction is rewarded and encouraged to be visited more.”

Passive voice makes this hard to parse. The agent is the one making predictions and receiving rewards.

“understand our audio-driven exploration works well under what circumstances”

Some words are missing here

“can encourage interest action that involved physical interaction”

Needs fixing

- Related Work

“By leveraging audio-visual correspondences in videos, it can help to learn powerful audio and visual representations through self-supervised learning”

It can help to learn? -> It can learn?

“Reinforcement Learning (RL)”

Acronym has already been introduced.

“makes use of the bootstraps for deep exploration”

bootstraps -> bootstrap

“Here, we mainly focus on the problem of using intrinsic rewards to drive explorations.”

Why is exploration plural?

“The most widely used intrinsic motivation could be roughly divided into two families.” could -> can

The works discussed in the ensuing sentences are already listed and cited in the previous sentences. If you are going to discuss as if you had not already just mentioned them, it is better to exclude them from the previous sentences.
Also, maybe you should add “approaches” or “methods” somewhere in this sentence.

“Burda et al. (2018b) employs the prediction errors of a self-state feature extracted from a fixed and random initialized network and encourage the agent to visit more previous unseen states.” previous -> previously

What is a self-state feature? Burda does not use that term and there is no explanation here.
The end of the sentence doesn’t make sense. What is the subject of “encourage”?

“Another one is the curiosity-based approach (Stadie et al., 2015; Pathak et al., 2017; Haber et al., 2018; Burda et al., 2018a), which is formulated as the uncertainty in predicting the consequences of the agent’s actions.”

-The paper previously said that there were two families so “another one” should be “The other” or something else signifying that this is referring back to the two families comment.

-The paper described family one as a set of approaches (plural) but family two as an approach “singular”.

-The end of the sentence needs fixing.

“The agent is then encouraged to improve its knowledge about the environment dynamics” then -> thereby

Figure 1 would be more clear if it did not use time indexing for the stage 1 section.

“There are numerous works to explore”

that explore?

“More recently, Dhiraj et al. (2020) collected a large sound-action-vision dataset using Tilt-bolt and demonstrates sound signals could provide valuable information for find-grained object recognition”

-Dhiraj et al. is plural so demonstrates should be demonstrate.

-could provide -> can provide

-find-grained -> fine-grained

“More related to us” us -> our work OR this work

“which have shown that the sound signals could provide useful supervisions for imitation learning and reinforcement learning in Atari games.”

-could -> can

-unnecessary passive voice


Throughout the Sounds and Actions paragraph, the paper repeatedly switches between describing papers in past tense and present tense.

“And then we elaborate on the pipeline of self-supervised exploration through auditory event predictions.” And then -> Then

There is nothing wrong (in general) with starting a sentence with the word “And” but this is not an appropriate place for it.

“a standard Markov Decision Process (MDP), defined as (S, A, r, T , µ, γ). S, A and µ(s) : S → [0, 1] denote”

as -> by

It is bad form to start a sentence with a symbol.

“The transition function T (s 0 |s, a) : S × A × S → [0, 1] defines the transition probability to next-step state s’ if the agent takes action a at current state s.”

The transition function defines this transition probability whether or not the agent executes a at s.

“The goal of training reinforcement learning is to learn an optimal policy π ∗ that can maximize the expected rewards under the discount factor γ as”

This is the goal of reinforcement learning not the goal of training reinforcement learning.

The sentence doesn’t work. It reads “The goal is to learn an optimal policy that is an optimal policy.” Either modify to “The goal is to learn an optimal that is an optimal policy.” or “The goal is to learn an optimal policy. An optimal policy is …”

“The agent chooses an action a from a policy π(a|s)” from -> according to

“Intrinsic Rewards for Exploration”

 This paragraph is repeating information that was already written

“Designing intrinsic rewards for exploration has been widely used to resolve the sparse reward issues in the deep RL communities”

-Unnecessary passive voice

-What deep RL communities? Isn’t there just one? If there are more than one, what are they?

“transits to the next state with visual observation sv,t+1 and sound effect ss,t+1. “

transits -> transitions

state with -> state, receiving

“We hypothesize that the agents, through this process, could learn the underlying causal structure of the physical world and use that to make predictions about what will happen next, and as well as plan actions to achieve their goals.”

-“and as well as” is redundant

“To better capture the statistic of the raw auditory data”

What statistic?

“For the task of auditory event predictions, perhaps the most straightforward option is to directly regress the sound features Φ(ss,t+1) given the feature embeddings of the image observation sv,t and agent’s actions at”

The paper said this already

“Nevertheless, we find that not very effective. We hypothesize that the main reasons are: 1) the mean squared error (MSE) loss used for regression is satisfied with “blurry” predictions. This might not capture the full distribution over possible sounds and a categorical distribution over clusters; 2) the MSE loss does not accurately reflect how well an agent understands these auditory events. Therefore, we choose instead to define explicit auditory events categories and formulate this auditory event prediction problem as a classification task, similar to (Owens et al., 2016b).”

-Nevertheless is not appropriate here.

-The paper runs this experiment and discusses the results later in the paper. Its confusing to discuss it both as a choice and as an ablation.


“Our AEP framework”

While the acronym can be deduced from the section header, it is still good practice to write it explicitly.

“We need to”

This is a design choice, not a need.

“And then”

And is not appropriate here.

“takes input as the embedding of visual observation and action and predicts which auditory event will happen next.”

needs fixing

“then utilized” -> used

“to explore those auditory events with more uncertainty”

those is unnecessary here

“We will elaborate on the details of these two phases below.”

Get rid of “will”

“The agents start to collect audio data by interacting with the environment”

They start to isn’t good phrasing here.

“During this exploration, the number of clusters will grow”

fix tense

“After the number of the clusters is saturated”

The description of the sound clustering process is muddled. It does not defined saturated until after it has used it in context.

“To be noted, the number of cluster K is determined automatically in our experiments. In practice, we define K ∈ [5, 30] for clustering at each time step and use the silhouette score to automatically decide the best K, which is a measure of how similar a sound embedding to its own cluster compared to other clusters.”

Do “To be noted” or “In practice” have any added value here?

“we believe it is “saturated””

The paper is defining saturation, it is not a belief.

“We visualize the corresponding visual states in two games (Frostbite and Assault) that belong to the same sound clusters, and it can be observed that each cluster always contains identical or similar auditory events”

-Unnecessary passive voice

-Always is a pretty strong claim. Can we deduce that from this visualization?

“Since we have already explicitly defined the auditory event categories, the prediction problem can then be easily formulated as a classification task”

-Don’t need “already”

-Don’t need “then”

“It will reward the agent at that stage and encourage it to visit more such states since it is uncertain about this scenario”

-Use present tense tense

-at that state?

“In practice, we do find that the agent can learn to avoid dying scenarios in the games since that gives a similar sound effect it has already encountered many times and can predict very well.”

-we do find -> we find

-dying scenarios?

- Experiments

“And then”

And is not appropriate here

“Our primary goal is to investigate whether we could use auditory event prediction as intrinsic rewards to help RL exploration.”

This statement is repeated many times in the paper and has no added value to this section.

“also supports an audio API to provide”

to provide -> that provides

“We use 20 familiar video games”

Is familiar needed here?

“We follow the standard setting in (Pathak et al., 2017; Burda et al., 2018b), where an agent can use external rewards as an evaluation metric to quantify the performance of the exploration strategy. This is because doing well on extrinsic rewards usually requires having explored thoroughly.”

-Passive voice is excessive in multiple places here

-incorrect use of interrogatives

“the the” typo

“Table 1: Categorical results”

These are not results

“could simulate”

can simulate

“physic simulation platform” physics

“Figure 7 ,”comma misplaced

“The agent is required to execute actions to interact with objects of different materials and shapes.”

The agent interacts with objects of different materials and shapes.

“When two objects collide, the environment could generate collision sound based on the physical properties of objects”

could generate collision sound?

“We would like to compare”

We compare

“we choose PPO algorithm”

either choose the PPO algorithm or choose PPO

“The PyTorch implementation”

which?

“the open-source toolbox 1”

-which?

-footnote shouldn’t have space

“As for the auditory prediction network”

Don’t need “as”

“our model use 10K interactions” fix

“previous vision-only intrinsic motivation modules”

don’t need previous here

“We would like to provide an in-depth understanding of under what circumstances our algorithm works well.”

You would like to or you do?

“event-driven sounds which emitted when agents” fix

“action-driven sounds which emitted when agents” fix

“None of the category accounts for the majority” fix

“Since the sound is more observable effects of action” fix

“games dominant with event-driven” fix

“These are also reasonable”

Nothing for “These” to refer to

“Sometimes sound events will occur independently of the agent’s decisions and do not differentiate between different policies”

differentiate is not the right word here

The phrase “ablated study” is used repeatedly. Use “ablation study” instead?

“We further carry out additional experiments”

further unnecessary

“One main contribution of our paper is to use auditory event prediction as an intrinsic reward.”

Repeated without added value

“In Figure 6, using only 16K exploration steps, our agent has already explored all unique states (211 states)” fix

“These results demonstrate that our agent explores environments more quickly and fully, showing the potential ability of exploring the real world.”

Do they? This is a strong claim

“less than 195 unique states” fewer

“These results demonstrate that our agent explores environments more quickly and fully, showing the potential ability of exploring the real world.”

Do they? Again, a strong claim.

There is a sub-header “Setup” of Section 4.4 Experiments on TDW. There is also a TDW sub-header of Section 4.1 Setup.

“The action space consists of moving to eight directions and stop. An action is repeated 4 times on each frame.”fix

“previous vision-based modules”

don’t need previous

“could not” -> did not

“the 3D photo-realistic world, in which a physical event happens.”needs fix

“Instead, our auditory event prediction driven exploration will lead agents”

In constrast, … exploration leads agents

“(See SHE in Figure 8)” see

“We will reward an agent” We reward an agent

“We also want to understand if it is necessary to use audios” audios?

“is powerful for agents to build a causal model of the physical world” fix

- Comments on Organization

-I think it would make more sense to group the question-based analysis together. IE, the ablation studies and the discussions.

-I think whether to use clustering classification or feature prediction should be discussed consistently throughout the paper. As it currently stands, sometimes it is presented as a design choice, sometimes with the claim that the latter is worse (without supporting evidence), and sometimes as a question to be answered by an ablation study.

- Other comments

“As compared to visual cues, sounds are often more directly or easily observable causal effects of actions and interactions.”

Is this obvious?

# Causality?

The paper makes a number of comments about its method learning causal structure. To me, these seem like big claims. The proposed algorithm has no mechanism that tests counterfactuals or, as far as I can tell, any other mechanism for estimating causation, so I see no reason why it would learn anything beyond correlative relationships. Given this fact, in my opinion, if the paper wants to make claims about the fact that its method is learning causal structure, it should back these claims up with experiments.

# Experimental Evaluation

The paper makes very definitive claims (see writing section) about the effectiveness of its method compared to the baselines. In my opinion, there are a number of issues with these claims. First, the paper gives no information (that I could find) about how it tuned the baselines or whether they tuned them at all. Second, for the Atari results, the paper states that it used three random seeds. For the other experiments, it does not say how many seeds it used (or at least I could not find where it said so). Both of these facts make it difficult to know how seriously to take the claims of superior performance.


# Related work

The paper cites many references in its related work section. Yet, I feel that it tells us almost nothing about what it is most important for us to know about:

"More related to us are the papers from Aytar et al. (2018) and Omidshafiei et al. (2018), which have shown that the sound signals could provide useful supervisions for imitation learning and reinforcement learning in Atari games. Concurrent to our work, Dean et al. (2020) uses novel associations of audio and visual signals as intrinsic rewards to guide RL exploration. Different from them, we use auditory event predictions as intrinsic rewards to drive RL explorations."

We are only given a couple of sentences of information about these papers. Additionally, I am not sure that is consistent that the paper claims that Dean is concurrent, but at the same time, designed its experiments to follow Dean “Following (Dean et al., 2020), we use the the apartment 0 in Replica scene (Straub et al., 2019) with the Habitat simulator for experiments.”

# Choice of baselines

The paper appears to be concerned with claiming superior performance over vision-based intrinsic methods, yet it is not clear to me that having superior performance is necessary for the proposed method to have added value, given that the two methods make use of disjointed information for prediction targets. I do not mean to say that these experiments do not have added value—certainly it is nice to see the proposed algorithm compared to existing algorithms—but maybe the adversarial narrative is not the right choice? I understand that the paper is claiming that sounds give the agent better information for intrinsic exploration but, in my opinion, convincing evidence for that claim would require extensive experiments on a wide array of intrinsic methods using both sound and vision and many environments.

# Broader Scope Question

Can a paradigm requiring an initial exploration stage to collect diverse sounds be effective in environments in which some sound events require a large amount learning to discover?

# Closing Thoughts

This seems like promising work, but in my opinion it is not ready for publication. Both the quality of writing and the issues with seeds/tuning independently merit rejection. There are also other issues discussed above.

---

### Author Response · Authors · 2020-11-21
**[Pre-revision] General Response**

We thank all reviewers for their constructive and insightful suggestions to strengthen this work. We are also glad that all the reviewers think our idea is interesting and novel. In addition to the specific response below, here we summarize our goals, address some common concerns, and describe the changes planned to be included in the revision.


### Our Goal:

In this paper, we propose a novel module using auditory events prediction as an intrinsic module to drive RL exploration without extrinsic rewards.

### Our Achievement:

1. We perform an in-depth study on 20 Atari game environments to understand the potential and limitations of our models (See Table 1).

2. We show our new module could improve audio-visual exploration in the Habitat, a photo-realistic virtual house navigation environment.

3. We demonstrate that our module could collect more physical interaction in the TDW, a high-fidelity multi-modal physical simulation environment.


### Common concerns:


> Experimental setting on Habitat.

We follow the audio-visual navigation setting used [1,2]  by placing a fixed audio clip to a fixed location.

The audio engine in Habitat could simulate various sound waveforms at arbitrary agent receiver positions based on the room’s geometry, major structures, and materials. Even though there is only one sound source, the agent hears differently in each location.  The auditory observation is determined jointly by the agent's position and the sound source.   So the agent could leverage the discovered latent auditory events to drive exploration.


> Reasons for using sound texture.

   We use the open-source code to extract the sound texture features。

   https://github.com/andrewowens/multisensory/blob/master/src/aolib/subband.py


  The reasons we use sound textures are two-fold:
- Extracting sound texture is more computationally efficient compared with other deep learning-based methods.
- Sound textures have been successfully used in prior work on audio-visual learning [3,4]. We use their implementations for the new audio-visual RL exploration application.


> Differences compared with [5,6]
- The primary goal of our work is to design an auditory event prediction based intrinsic module to drive RL explorations without extrinsic rewards.
- In [5], they propose to use multi-sensory for training RL with extrinsic rewards.
- In [6],  they propose to use multi-sensory for imitation learning.

- We have adapted their baseline to our problem setup. The results could be found in section B of supplementary materials.

### Planned changes:

1.  We will add a baseline using binary sound event classification as an intrinsic reward.

2.  We will add a baseline using audio-visual correspondences [1] as an intrinsic reward.

3.  We will add a baseline using visual data to define the events.

4.  We will add an ablated study using RND to collect 10K interaction data.

5.  We will give more explanations on the relationship between auditory event prediction and the causal effects of actions.

6.  We will carefully proofread the typos.

[1] Victoria Dean, Shubham Tulsiani, Abhinav Gupta. See, Hear, Explore: Curiosity via Audio-Visual Association. https://arxiv.org/abs/1912.11474

[2] Changan Chen, Unnat Jain, Carl Schissler, Sebastia Vicenc Amengual Gari, Ziad Al-Halah, Vamsi Krishna Ithapu, Philip Robinson, Kristen Grauman. SoundSpaces: Audio-Visual Navigation in 3D Environments. ECCV 2020

[3] Andrew Owens, Jiajun Wu, Josh McDermott, William T. Freeman, Antonio Torralba. Ambient Sound Provides Supervision for Visual Learning. ECCV 2016

[4] Andrew Owens, Phillip Isola, Josh McDermott, Antonio Torralba, Edward H. Adelson, William T. Freeman. Visually Indicated Sounds. CVPR 2016

[5] Shayegan Omidshafiei, Dong-Ki Kim, Jason Pazis, Jonathan P. How. Crossmodal Attentive Skill Learner. AAMAS 2018.

[6] Yusuf Aytar, Tobias Pfaff, David Budden, Tom Le Paine, Ziyu Wang, Nando de Freitas. Playing hard exploration games by watching YouTube. NIPS 18

---

### Author Response · Authors · 2020-11-22
**Individual responses are updated, and look forward to reviewers' feedback!**

Dear reviewers:

We thank all reviewers for their constructive comments. This is one of the first works to use audio as intrinsic rewards to drive RL exploration. We are glad to see all reviewers agree our idea is novel and interesting. We also agree with the reviewers that it’s important to have more ablated studies to understand our model. We have listed all the planned changes in our general response above. Please don’t hesitate to let us know of any additional comments on the paper or on the planned changes.

---

### Author Response · Authors · 2020-11-24
**General Response: Revision Updated**

We would like to thank the reviewers for their thoughtful feedback. We are glad to see that reviewers generally appreciated the contributions of our paper – the novel idea of using auditory events to drive exploration (R1, R2, R3, R4, R5), the integration of multisensory information (R2), the motivation (R3), the comprehensive experiment environments (R1, R2), the analysis as well as the ablative analysis (R1), and the writing clarity (R5).

We would like to emphasize again that our main contributions are:

- We introduce a novel and effective auditory event prediction (AEP) framework to make use of the auditory signals as intrinsic rewards for RL exploration.

- We perform an in-depth study using 20 Atari games to understand our audio-driven exploration module.

- We demonstrate our new model can enable a more efficient exploration strategy for audio-visual embodied navigation on the Habitat environment.

- We show that our new intrinsic module is more stable in the 3D multi-modal physical world environment and can encourage interest actions that involved physical interactions.

We have revised our manuscript to include the following changes:

- We have included a new baseline using binary sound event classification as an intrinsic reward in Sec. 4.5.

- We have included a new baseline using audio-visual correspondences as an intrinsic reward in Sec. 4.5.

- We have included a new baseline using visual data to define the events. in Sec. 4.5.

- We have included an ablated study and reported the performance by collecting audio data via RND in Figure 5.

- We have added more discussion on the limitation of regressing feature embedding in Sec. 3.2

- We have provided more details on the experimental setup in the Habitat environment in  Sec. 4.3.

- We have provided more details on the sound-clustering baseline in  Sec. 4.2.

- We have provided more explanations on using sound textures as audio representations in  Sec. 3.2.

Please don't hesitate to let us know of any additional comments on the manuscript or the changes.

---

### Decision · Program_Chairs · 2021-01-07
**Final Decision**

**Decision:**

Reject

**Comment:**

Investigating using other sensory inputs in our agents, and the impact on exploration is fascinating. We all want to see agents that use more sensory information.

As it stands the paper has several issues that require significant revision, most notably: (1) the polish, quality of the writing and clarity of the text is low, (2) the empirical results are based on 3 runs---at this number we might not have enough data to form valid estimates of the std dev---the error bars are not defined (see Henderson et al 2018), (3) in the ablation studies the hyper-parameters are not tuned (as far as the text suggests) meaning the ablations results might be not representative of the utility of the method, (4) many missing details like hyper-parameter tuning, number of runs in some cases, and reasonable descriptions of experiment protocols and baselines, (5) unsupported claims of causality.

Some of the issues were first raised during the discussion period, so another reviewer was brought in and provided a high quality review with many constructive comments. All reviewers reached clear agreement at the end of the discussion period.